# Reliability of Systematic and Targeted Biopsies versus Prostatectomy

**DOI:** 10.3390/bioengineering10121395

**Published:** 2023-12-06

**Authors:** Tianyuan Guan, Abhinav Sidana, Marepalli B. Rao

**Affiliations:** 1College of Public Health, Kent State University, Kent, OH 44240, USA; 2Division of the Biological Sciences, The University of Chicago, 5841 S Maryland Avenue, Chicago, IL 60637, USA; abhinavsidana@uchicago.edu; 3Division of Biostatistics and Bioinformatics, University of Cincinnati, Cincinnati, OH 45219, USA; marepalli.rao@uc.edu

**Keywords:** prostate cancer, systematic biopsy, targeted biopsy, ROC curve, area under the curve, sensitivity, specificity, logistic regression, biomarker, machine learning methods

## Abstract

Systematic Biopsy (SBx) has been and continues to be the standard staple for detecting prostate cancer. The more expensive MRI guided biopsy (MRITBx) is a better way of detecting cancer. The prostatectomy can provide an accurate condition of the prostate. The goal is to assess how reliable SBx and MRITBx are vis à vis prostatectomy. Graded Gleason scores are used for comparison. Cohen’s Kappa index and logistic regression after binarization of the graded Gleason scores are some of the methods used to achieve our goals. Machine learning methods, such as classification trees, are employed to improve predictability clinically. The Cohen’s Kappa index is 0.31 for SBx versus prostatectomy, which means a fair agreement. The index is 0.34 for MRITBx versus prostatectomy, which again means a fair agreement. A direct comparison of SBx versus prostatectomy via binarized graded scores gives sensitivity 0.83 and specificity 0.50. On the other hand, a direct comparison of MRITBx versus prostatectomy gives sensitivity 0.78 and specificity 0.67, putting MRITBx on a higher level of accuracy. The SBx and MRITBx do not yet match the findings of prostatectomy completely, but they are useful. We have developed new biomarkers, considering other pieces of information from the patients, to improve the accuracy of SBx and MRITBx. From a clinical point of view, we provide a prediction model for prostatectomy Gleason grades using classification tree methodology.

## 1. Introduction

Detection of cancer in the prostate gland is fraught with difficulties. The character of prostate cancer is different from cancers in other organs. Some segments of the prostate are cancerous, other segments benign, and the rest metastatic. Systematic biopsy (SBx), in which needles are inserted into the prostate to extract tissue, is commonly used to obtain information from the prostate in the form of a Gleason score. The biopsy may completely miss the cancerous part of the prostate. Multiparametric magnetic resonance imaging (mpMRI) targeted biopsy (MRITBx) is being accepted as a more reliable screening test for the detection of cancer [1,2,3]. The needles are guided using MRI. The test could miss cancer. There is a need to assess how reliable these biopsies are. For assessing reliability, one needs a definitive procedure. It is prostatectomy, in which the prostate is removed and examined. However, prostatectomy is not a gold standard. If the prostate is removed, there is no gland left to treat. It will be a boon if one has data on patients with information on both SBx and prostatectomy, and MRITBx and prostatectomy. We are bestowed with such a boon. 

There are research papers proposing new ways of detecting prostate cancer [4,5,6,7,8,9,10,11,12], most of them using machine learning methods. We do not know how reliable these methods of detection are. They do not have the benefit of definitive procedures of detection to compare with. 

One cannot see how one can develop a gold standard procedure without removing the prostate. Of course, removing the prostate is a treatment which is not common. There is a gap in how to examine the reliability of any detection procedure vis-à-vis prostatectomy. We are filling this gap by focusing on the biopsies, SBx and MRITBx. The department of Urology at the University of Cincinnati has been collecting data on patients who came for prostate screening for many, many, years. One segment of the data has information on the results of SBx, MRITBx, and prostatectomy. This is our core data for the assessment of the screening tests vis-à-vis prostatectomy.

A summary of acronyms is provided below:

SBx: Systematic Biopsy

MRITBx: Multiparametric Magnetic Resonance Imaging Targeted Biopsy

PSA: Prostate-specific Antigen

DRE: Digital Rectal Examination

ROC: Receiver Operating Characteristic

HCsys: The Systematic Biopsy Gleason Grades binarized, analogously, Grades 1, 2, 

3 versus Grades 4, 5

HCpros: The prostatectomy Biopsy Gleason Grades binarized, analogously,

Grades 1, 2, 3 versus Grades 4, 5 

HCTa: The Targeted Biopsy Gleason Grades binarized, analogously, with Grades 

1, 2, 3 (0) versus Grades 4, 5

## 2. Materials and Methods

Our study was approved by the University of Cincinnati institutional review board (UC IRB: 2018-4010). The data have information on Gleason scores from patients on SBx, MRITBx, prostatectomy and many other covariates. Besides demographic details, the data have information on PSA (prostate-specific antigen), prostate volume, DRE (digital rectal examination), and family history. We performed a retrospective study of patients with newly diagnosed status of prostate cancer at UC Health between September 2014 and April 2020. The final cohort had 597 patients for analysis. For our analysis, we included those patients with data on prostatectomy, SBx, and MRITBx. The final size of one of the data sets is 235, with information on both SBx and prostatectomy results along with covariates. The other data set has a size of 104, with information on both MRITBx and prostatectomy along with covariates.

Patient demographic, clinical, and pathological data were recorded. The Gleason scores are categorized into 5 Grades (1, 2, 3, 4, 5) [13], as described in Table 1. 

The workflow of our research is presented below:Grade 1, 2, 3, 4 or 5, as per SBx in comparison to Grade 1, 2, 3, 4 or 5 as per prostatectomy (Cohen’s Kappa)Grade 1, 2, 3, 4 or 5, as per MRITBx in comparison to Grade 1, 2, 3, 4 or 5 as per prostatectomy (Cohen’s Kappa)Grade 1 vs. Grade 2, 3, 4 or 5 as per SBx in comparison to Grade 1 vs. Grade 2, 3, 4 or 5 as per prostatectomy (Logistic Regression)Grade 1 vs. Grade 2, 3, 4 or 5 as per MRITBx in comparison to Grade 1 vs. Grade 2, 3, 4, 5 as per prostatectomy (Logistic Regression)Grade 1 or 2 vs. Grade 3, 4, 5 as per SBx in comparison to Grade 1 or 2 vs. Grade 3, 4, 5 as per prostatectomy (Logistic Regression)Grade 1 or 2 vs. Grade 3, 4, 5 as per MRITBx in comparison to Grade 1 or 2 vs. Grade 3, 4, 5 as per prostatectomy (Logistic Regression)Grade 1 or 2 vs. Grade 3, 4, 5 as per MRITBx in comparison to Grade 1 or 2 vs. Grade 3, 4, 5 as per prostatectomy (Classification Tree)

Significant cancer was defined as a Gleason score ≥ 7. Several methods were employed to contrast the Gleason grades as per prostatectomy and SBx on one hand, and prostatectomy and MRITBx on the other hand. The Kappa statistic [14,15] is calculated to assess the degree of agreement between the Gleason grades of SBx and prostatectomy, MRITBx and prostatectomy. For other contrasts, the Gleason grades are binarized in several different ways. At each type of binarization, SBx is assessed vis-à-vis prostatectomy in terms of sensitivity and specificity. In a similar way, MRITBx is assessed vis-à-vis prostatectomy. A logistic regression model is fitted to each of the binarized Gleason grades of prostatectomy with several predictors, including the corresponding binarized Gleason grades of SBx. We then developed a biomarker out of the logistic regression model and assessed its utility for prediction by calculating the area under the ROC (Receiver Operating Characteristic) curve of the biomarker. The Youden method [16,17] is used to put forward a diagnostic (screening) test. The sensitivity and specificity of the diagnostic test are calculated to assess the effectiveness of the diagnostic test. An identical methodology is used for binarized Gleason grades of prostatectomy, with several predictors including the corresponding binarized Gleason grades of MRITBx. 

Following Table 1, there are two ways to binarize the Gleason grades. One simple and natural way is to take the binarized levels to be A = {1} and B = {2, 3, 4, 5}. Another way is to take A = {1, 2, 3} and B = {4, 5} [13]. The binarization allowed us to fit logistic regression models with the two ways of binarizing prostatectomy Gleason grades. In each model, we take the binarized prostatectomy Gleason grades as the outcome variable and the corresponding binarized SBx Gleason grades as the principal predictor. In the first method of binarization, there are only 4 cases in the A = {1} group. The logistic regression model fitting is not advisable [18,19]. Therefore, we focused on the binarization of the Gleason grades into A = {1, 2, 3} and B = {4, 5}. We have employed the cross-validation method (LOOCV: leave one out cross-validation) on the model comparing high cancers vs. not high cancers using the predictors age, race, prostate volume, PSA, and high cancers vs. not high cancers, as per SBx. In the same context, we have employed the K-fold cross-validation method. A screening marker is developed from the regression model and its utility is assessed for the screening test by its ROC curve. A screening test is laid out with a cut point determined by the Youden method [16,17], along with its sensitivity and specificity. Similar pursuits are carried out with MRITBx. Statistical analysis was performed by using the computing software R 4.3.0 (R Core Team, 2017) [20]. The Kruskal–Wallis rank sum test was used to compare the medians of continuous variables. Pearson’s Chi-squared test and Fisher’s exact test were used to compare proportions of categorical variables. A crowning achievement was to employ machine learning methods to develop a prediction model on three fronts. On one front, the response variable is taken to be the categorical variable, the Gleason grades of prostatectomy. On the other two fronts, the response variable is taken to be the binarized Gleason grades of prostatectomy, binarized in two different ways as enunciated above.

## 3. Results

The focus was on comparing the diagnoses stemming from SBx vs. prostatectomy on one hand, and MRITBx vs. prostatectomy on the other hand. The first step in the analysis was to make overall comparisons via the Kappa Statistic. The second step in the analysis was to make comparisons with respect to cancer vs. no cancer. The third step in the analysis was to make comparisons with respect to high cancers vs. not high cancers. Ultimately, we developed biomarkers to discriminate high cancers vs. not high cancers built on SBx and MRITBx by including additional predictors. We showed that these biomarkers are highly accurate, with more than 90% accuracy. The key methodology we used was the logistic regression model. The work was supplemented by the machine leaning method, classification tree. 

### 3.1. Kappa Statistic

The data covers the period from September 2014 to April 2020 with a cohort of 597 patients. An SBx Gleason score was determined for every patient in the study. The data reports Gleason scores: benign, 3 + 3, 3 + 4, 4 + 3, 4 + 4, 4 + 5, 5 + 4, and 5 + 5. The scores are categorized into 5 grades: benign or 3 + 3 = Grade 1; 3 + 4 = Grade 2; 4 + 3 = Grade 3; 4 + 4, 3 + 5 or 5 + 3 = Grade 4; 4 + 5, 5 + 5, or 5 + 4 = Grade 5. A prostatectomy was performed on only 235 patients. Only 104 patients received MRITBx. The following table (Table 2) shows the Gleason grades along with the frequencies.

This table is alarming. Suppose diagnosis was made based on SBx. As per the prostatectomy diagnosis, only 4 out of 235 fall into Grade 1. On the other hand, as per the SBx diagnosis, 41 out of 235 fall into Grade 1. For a substantial number of patients, cancer diagnosis was missed out by SBx.

Suppose the diagnosis was made based on MRITBx. As per the prostatectomy diagnosis, only 3 out of 104 fall into Grade 1. On the other hand, as per the MRITBx diagnosis, 24 out of 104 fall into Grade 1. For a substantial number of patients, cancer diagnosis was missed out by MRITBx.

We evaluated how close the agreement is between Gleason grades of prostatectomy and SBx overall. The Cohen’s Kappa [14,15] is 0.31, which means a fair agreement, with a 95% CI [0.23, 0.39]. For prostatectomy vs. MRITBx, the index is 0.34, which also means a fair agreement, with a 95% CI [0.23, 0.45]. On a binary level, a value of Kappa greater than 0.75 is considered as an excellent agreement, whereas lower than 0.4 is treated as a poor agreement [14,15].

A simple diagnostic test was developed following the data in Table 1. To discriminate the graded score ≥ 2 from the graded score = 1 truly, we used the following screening test. 

Test is positive if the graded score ≥ 2 under SBx.

Test is negative if the graded score = 1 under SBx.

To assess the effectiveness of the test versus prostatectomy, we used Table 3.

Sensitivity of the test = 192/231 = 0.83.

Specificity of the test = 2/4 = 0.50.

We can use MRITBx for a screening test to discriminate the graded score ≥ 2 from the graded score = 1. The relevant screening test was given by:

Test is positive if the graded score ≥ 2 under MRITBx.

Test is negative if the graded score = 1 under MRITBx.

To assess the effectiveness of the test versus prostatectomy, we used Table 4.

Sensitivity of the test = 79/101 = 0.78.

Specificity of the test = 2/3 = 0.67.

Overall, MRITBx is a better procedure compared with SBx. 

We embarked on improving the diagnostic test based on SBx by including some information on the patients. We fitted a logistic regression model with the response variable binarized prostatectomy (Level 1—prostatectomy positive: 2, 3, 4, 5; Level 2—prostatectomy negative: 1), and predictors: age, race, prostate volume, PSA, DRE and family history of prostate cancer. The number of prostatectomy negatives is only 4, which is less than 10% of the total size of the sample 235. Logistic regression for these binarized grades is not recommended [18,19]. We desisted including the results from this data analysis exercise. 

We binarized the grades in a different way: detect high/very high cancers from not high/very high cancers [21]. We fit a logistic regression model with the binarized response variable (Level 1—prostatectomy high/very high cancers: grades 4, 5 versus Level 2—prostatectomy not high/very high cancers: grades 1, 2, 3), and predictors: age, race, prostate volume, PSA, DRE and family history. We fitted two separate logistic regression models. In one, we included the corresponding binarized SBx Gleason grades. In the other, we included the corresponding binarized MRITBx Gleason grades. 

### 3.2. SBx Gleason Grades Binarized as a Predictor in the Model

Two sets of logistic regression models were run. In one set, the predictors were age, race, prostate volume, PSA, and HCsys (The systematic biopsy Gleason grades binarized, analogously, Grades 1, 2, 3 versus Grades 4, 5). The model fit was good with the ratio of residual deviance and degrees of freedom less than 1 (*p*-value = 1). The significant predictors were PSA (*p*-value = 0.0465) and Hcsys (*p*-value < 0.0001). The implication is that SBx coupled with PSA is a good predictor of the true condition (prostatectomy: Grades 1, 2, 3 vs. Grades 4, 5). The output is given in Appendix A. 

We developed a biomarker based on predictors to discriminate the levels of Hcpros. The biomarker is the logit of the model, i.e.,
$$Logit=-4.503+0.01*Age+0.999*Race\;\left(Caucasian\right)+\left(-0.127\right)*Race\;\left(Other\right)+\;\;2.982*HCsys+\left(-0.0033\right)*Volume+0.015*PSA$$

The Logit can be computed for a patient with information on age, race, Hcsys, prostate volume, and PSA. The result is the biomarker value of the patient. 

The race, Black, was the baseline of race. With the parameters of the model estimated, the logit was computable for everyone in the study. The summary statistics of the logit by the levels of Hcpros (1 = Grades 4, 5 and 0 = Grades 1, 2, 3) were tabulated in Table 5.

The logit values of the level 1 of Hcsys were generally higher than those of level 0. The Kernel density curves (Figure 1) attest to this phenomenon. This density curves indicate how good the biomarker logit is to discriminate high cancers vs. not high cancers.

In Figure 1, the density curve associated with level 1 (high/very high cancers) of Hcpros is on the right side of the curve associated with level 0 (not high/very high cancers). This was an indication that the biomarker would be a good discriminator of high/very high cancers versus not high/very high cancers. The ROC curve associated with the biomarker is given in Figure 2.

The area under the curve (AUC) was 87.5% with a 95% confidence interval 80.8% to 94.2%. The arrow pointed to our choice of the cut point −2.722 (Youden Method) with the specificity, 0.838 and the sensitivity, 0.833.

In view of Table 5, a diagnostic test for discriminating the levels of Hcpros has the following format.

Test is positive indicating high/very high cancers if biomarker ≥ c, Test is negative indicating not high/very high cancers if biomarker < c, for some c.

Our choice of c was governed by the following optimality principle. Minimize (1- −sensitivity_c_)^2^ + (1−specificity_c_)^2^ with respect to c. The number 1 in the expression refers to the sensitivity and specificity of SBx. For each choice of c, sensitivity_c_ and specificity_c_ are the sensitivity and specificity, respectively, associated with the cutpoint c. Our optimization foray gave us c = −2.722 with the sensitivity, 83.3% and the specificity, 83.8%. Thus the biomarker based on SBx, and other predictors were a better choice than the one based on SBx alone.

Diagnostic Test based on SBx

Test is positive (indicating high cancer) if


$$\begin{array}{cc} Logit= & -4.503+0.01*Age+0.999*\;Race\;\left(Caucasian\right)\\ & +\left(-0.127\right)*Race\;\left(Other\right)+2.982*HCsys\\ & +\left(-0.003\right)*Volume+0.015*PSA\;\geq -2.722 \end{array}$$


Test is negative (indicating not high cancer) if


$$\begin{array}{cc} Logit= & -4.503+0.01*Age+0.999*\;Race\;\left(Caucasian\right)\\ & +\left(-0.127\right)*Race\;\left(Other\right)+2.982*\mathrm{Hcsys}\\ & +\left(-0.003\right)*Volume+0.015*PSA\;<-2.722 \end{array}$$


### 3.3. MRITBx Gleason Grade Binarized as a Predictor in the Model

In another set, the predictors were age, race, prostate volume, PSA, and HCTa. The MRI targeted biopsy Gleason grades are binarized, HCTa (the targeted biopsy Gleason grades binarized, analogously, with Grades 1, 2, 3 (0) versus Grades 4, 5 (1). The model fit was good with the ratio of residual deviance and degrees of freedom less than 1 (*p*-value = 1). The significant predictors were PSA (*p*-value = 0.0229) and HCTa (*p*-value < 0.0001). The implication was that MRITBx, coupled with PSA, was a good predictor of the true condition (prostatectomy: Grades 1, 2, 3 vs. Grades 4, 5). The output is given in Appendix B. 

We developed a biomarker based on predictors to discriminate the levels of Hcpros. The biomarker is the logit of the model, i.e.,
$$Logit=-0.246+0.007*Age+0.196*Race\;\left(Caucasion\right)+1.873*Race\;\left(Other\right)+2.565*HCTa+\left(-0.01\right)*volume+0.143*PSA$$

The Race, Black, was the baseline of race. With the parameters of the model estimated, the logit was computable for everyone in the study. The summary statistics of logit by the levels of Hcpros (1 = Grades 4, 5 and 0 = Grades 1, 2, 3) were tabulated in Table 6.

The logit values of level 1 of HCpros were generally higher than those of level 0. The Kernel density curves (Figure 3) attest to this phenomenon.

In Figure 3, the density curve associated with level 1 of HCpros is on the right side of the curve associated with level 0. This is an indication that the biomarker will be a good discriminator of high/very high cancers versus not high/very high cancers. The ROC curve associated with the biomarker was given in Figure 4.

The area under the curve (AUC) is 0.922% with a 95% confidence interval 83% to 1. The arrow pointed to the choice of the cut point −2.204 with the specificity, 0.9759 and the sensitivity, 0.9. In view of Table 6, a diagnostic test for discriminating the levels of HCpros has the following format.

Test is positive indicating high/very high cancers if biomarker ≥ c; Test is negative indicating not high/very high cancers if biomarker < c, for some c. 

Our choice of c was governed by the following optimality principle. Minimize (1 −sensitivity_c_)^2^ + (1 −specificity_c_)^2^ with respect to c. The number 1 in the expression refers to the sensitivity and specificity of MRITBx. For each choice of c, sensitivity_c_ and specificity_c_ are the sensitivity and specificity, respectively, associated with the cutpoint c. Our optimization foray gave us c = −2.975 with a sensitivity, 93.3% and specificity, 42.6%. Thus, the biomarker based on MRITBx and other predictors was a better choice than the one based on MRITBx alone. Further, the biomarker based on MRITBx and other predictors was a better choice than the one based on SBx and other predictors.

Diagnostic test based on MRITBx

Test is positive (indicating high cancer) if 


$$\begin{array}{cc} Logit= & -0.246+0.007*Age+0.196*Race\;\left(Caucasion\right)\\ & +1.873*Race\;\left(Other\right)+2.565*HCTa\\ & +\left(-0.01\right)*volume+0.143*PSA\geq -2.975 \end{array}$$


Test is negative (indicating not high cancer) if


$$\begin{array}{cc} Logit= & -0.246+0.007*Age+0.196*Race\;\left(Caucasion\right)\\ & +1.873*Race\;\left(Other\right)+2.565*HCTa\\ & +\left(-0.01\right)*volume+0.143*PSA<-2.975 \end{array}$$


### 3.4. SBx as a Predictor in Classification Tree

We developed a classification tree with the outcome variable as the binarized prostatectomy Gleason grades with the levels A = {1, 2, 3} and B = {4, 5}. The SBx is binarized correspondingly as a predictor. Additional predictors are included in the tree. The tree is produced in Figure 5.

The tree is used as a prediction model. The tree has four terminal nodes. The prediction proceeds as follows: if systematic Gleason grade = 1, 2, or 3, classify the subject’s prostatectomy Gleason grade as 1, 2, or 3; if systematic Gleason grade = 4 or 5 and prostate volume less than 28, classify the subject’s prostatectomy Gleason grade as 4 or 5; if systematic Gleason grade = 4 or 5, prostate volume greater than or equal to 28, and PSA less than 14, classify the subject’s prostatectomy Gleason grade as 1, 2, or 3; if systematic Gleason grade = 4 or 5, prostate volume greater than or equal to 28, and PSA greater than or equal to 14, classify the subject’s prostatectomy Gleason grade as 4 or 5. The misclassification rate of the tree is 23/235 = 9.8% or the accuracy of the tree is 91.2%.

### 3.5. MRITBx as a Predictor in Classification Tree

We developed a classification tree with the outcome variable as the binarized prostatectomy Gleason grades with the levels A = {1, 2, 3} and B = {4, 5}. The MRITBx is binarized correspondingly as a predictor. Additional predictors are included in the tree. The tree is produced in Figure 6.

The tree is used as a prediction model. The tree has four terminal nodes. The prediction proceeds as follows: if PSA less than 18, classify the subject’s prostatectomy Gleason grade as 1, 2, or 3; if PSA greater than or equal to 18, and prostate volume less than 37, classify the subject’s prostatectomy Gleason grade as 1, 2 or 3; if PSA greater than or equal to 18, and prostate volume greater than or equal to 50, classify the subject’s prostatectomy Gleason grade as 1, 2 or 3; if PSA greater than or equal to 18, prostate volume greater than or equal to 37 and less than 50, the subject’s prostatectomy Gleason grade as 4 or 5. The misclassification rate of the tree is 25/235 = 10.6% or the accuracy of the tree is 89.4%. The predictor HCTa is not present in the tree at all. There is a reason behind this. The column HCTa has 131 missing values. Among the non-missing values, there are only 9 cases of high cancers, which constitutes less than 10% of the total number of subjects. In other words, the tree is built based on predictors, not including HCTa. This defeats our goal of making HCTa the main predictor.

## 4. Discussion

There are several studies devoted to diagnosis of prostate cancer by prostatectomy [19,20,21,22,23]. A number of studies compare the efficacy of systematic biopsy (SBx) with other types of biopsies [24,25,26,27], none of which is a gold standard. Prostatectomy is accurate but cannot be a gold standard procedure. We have data on patients with information from SBx, MRITBx, and prostatectomy. This data provides a way to examine the efficacy of SBx vis-à-vis prostatectomy and that of MRITBx vis-à-vis prostatectomy. Such data enable us to develop a biomarker to discriminate high risk cancer (Grades 4, 5) and not high risk cancer (Grades 1, 2, 3). We showed that SBx with the predictor PSA is a better discriminator of high cancers and not high cancers, with the area under the ROC curve 87.5%, the sensitivity, 83.3% and the specificity, 83.8%, than the one just based on SBx alone (Cohen’s Kappa = 0.34, sensitivity = 73.3% and specificity = 87.3%). We showed that MRITBx with the predictor PSA is a better discriminator of high cancers and not high cancers with the area under the ROC curve 92.6%, the sensitivity, 93.3% and the specificity, 42.6%, than the one just based on MRITBx alone (Cohen’s Kappa = 0.16, sensitivity = 77.8% and specificity = 85.3%).

When prostatectomy Gleason grades and SBx Gleason grades are binarized with A = {1, 2, 3} and B = {4, 5}. The accuracy of the classification tree for discriminating high cancers and not high cancers is 91.2% when binarized systematic Gleason grades, PSA, and prostate volume were used as predictors. When prostatectomy Gleason grades and targeted Gleason grades are binarized with A = {1, 2, 3} and B = {4, 5}. The accuracy of the classification tree for discriminating high cancers and not high cancers is 89.4% when binarized targeted Gleason grades, PSA, and prostate volume were used as predictors. However, the targeted Gleason grades are not present in the tree because over 55% of its data is missing. The tree in Figure 5 is built on the predictors PSA and prostate volume. Some limitations in our study: when prostatectomy Gleason grades and SBx Gleason grades are binarized with A = {1} and B = {2, 3, 4, 5}, logistic regression and classification tree fail to explain prostatectomy Gleason grades because there are too few cases of A = {1}. The output is not reliable because there are only 4 cases with A = {1} among the prostatectomy grades. For the validity of logistic regression model, the frequency of A = {1} or B = {2, 3, 4, 5} should be at least 10% of the data [28,29].

To assess the accuracy of SBx and MRITBx, we need substantial data in each of the prostatectomy Gleason grades. The current data spanned January 2014 to March 2020. We are accumulating data from March 2020 onwards. We hope to have comprehensive data in the future to be able to assess accuracies.

## 5. Conclusions

This is the first time the efficacy of SBx vis-à-vis prostatectomy and that of MRITBx vis-à-vis prostatectomy were examined. We have developed biomarkers to discriminate high cancers and not high cancers using six-year clinical records data from the Urology Department at UCHealth. From our analysis, discriminating high cancers and not high cancers based on SBx by the logistic regression model, as well as the classification tree paradigm has an accuracy around 90% (Figure 5 and Figure 6 on Classification Tree). MRITBx has better accuracy compared with SBx when we use a logistic regression model (Figure 2 and Figure 4 on AUC). The models take information from additional predictors besides SBx and MRITBx. There are some limitations to our study. In the first place, our study used data from a single institution only. More trustworthy conclusions could be drawn from a multi-institutional study. Secondly, the potential for selection bias and a possible lack of powered analysis associated with the retrospective nature of the study must be noted. Lastly, we do not have enough data to discriminate cancer (Gleason Grade 2, 3, 4, 5) versus low cancer (Gleason Grade 1). Finally, we showed that the biomarker based on SBx and other predictors is a better discriminator of high cancers versus not high cancers than the one based on SBx alone. A similar conclusion holds for the biomarker based on MRITBx and other predictors.

The following are the biomarkers for detecting high cancers versus not high cancers. The sensitivities and specificities are reported along with the areas under their ROC curves. 

Biomarker based on SBx:$$\;Logit=-4.503+0.01*Age+0.999*Race\;\left(Caucasian\right)+\left(-0.127\right)*Race\;\left(Other\right)+\;\;2.982*HCsys+\left(-0.0033\right)*Volume+0.015*PSA$$

With specificity is 0.838 and sensitivity is 0.833.

Biomarker based on MRITBx:$$Logit=-0.246+0.007*Age+0.196*Race\;\left(Caucasion\right)+1.873*Race\;\left(Other\right)+2.565*HCTa+\left(-0.01\right)*volume+0.143*PSA$$
with specificity is 0.9759 and sensitivity is 0.9.

The diagnostic procedures based on SBx and MRITBx are not reliable for detecting cancer vs. no cancer. However, the procedures are excellent in detecting high cancer vs. not high cancer. The logistic regression model contrasting high cancers vs. not high cancers based on age, race, prostate volume, PSA and high cancers vs. not high cancers as per SBx has an accuracy of 84%. The cross-validation method as per LOOCV corroborated the accuracy with its own accuracy calculation at 86%.The K-fold cross-validation method put accuracy at 87% [30]. This is the main message coming from our paper.

If the high cancer determination is based on biopsies, the current practice is that high cancer is present if the Gleason Score is greater than or equal to 8. We have pointed out that this is not a good judgment. We can improve the diagnosis if we take into account age, race, prostate volume, and PSA.

Some recent literature on cancer detection has focused on machine learning methods [31,32]. From our perspective, we would like to assess how good these methods are vis-à-vis prostatectomy, if only we have data.

## Figures and Tables

**Figure 1 bioengineering-10-01395-f001:**
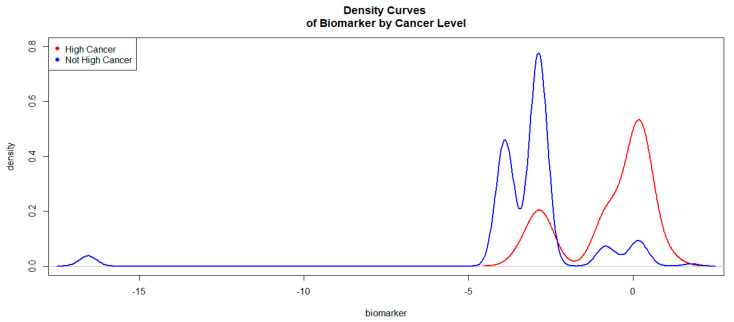
Kernel Density Curves of the Biomarker (based on SBx and other predictors) by Cancer Levels.

**Figure 2 bioengineering-10-01395-f002:**
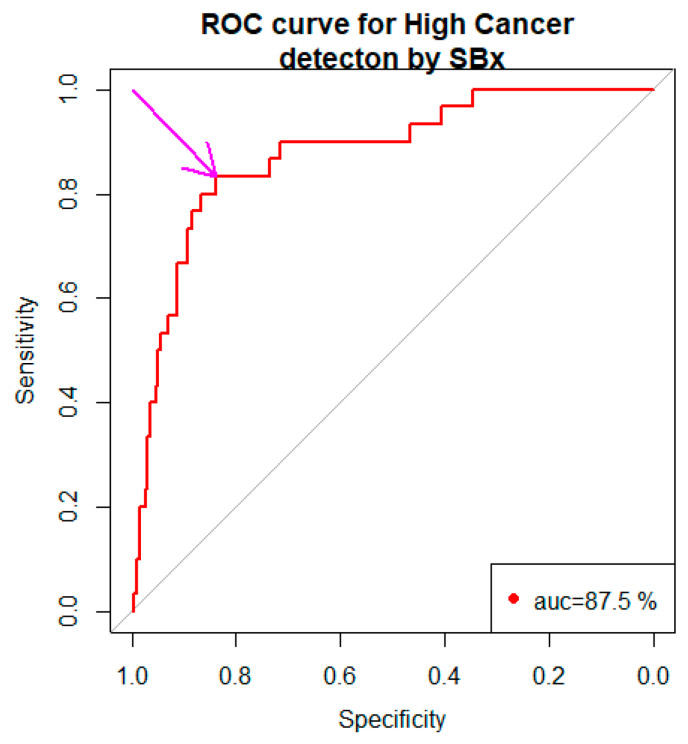
ROC curve of the biomarker (based on SBx and other predictors).

**Figure 3 bioengineering-10-01395-f003:**
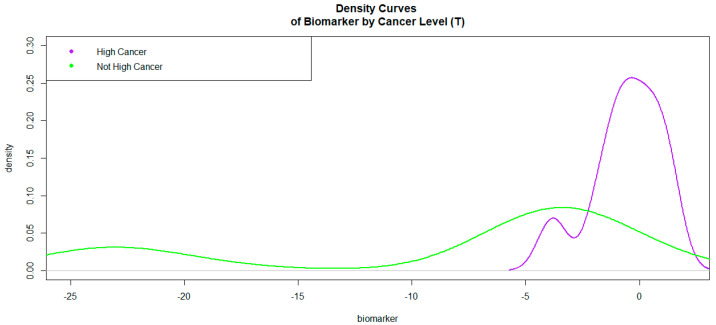
Kernel Density Curves of the Biomarker (based on MRITBx and other predictors) by Cancer Levels.

**Figure 4 bioengineering-10-01395-f004:**
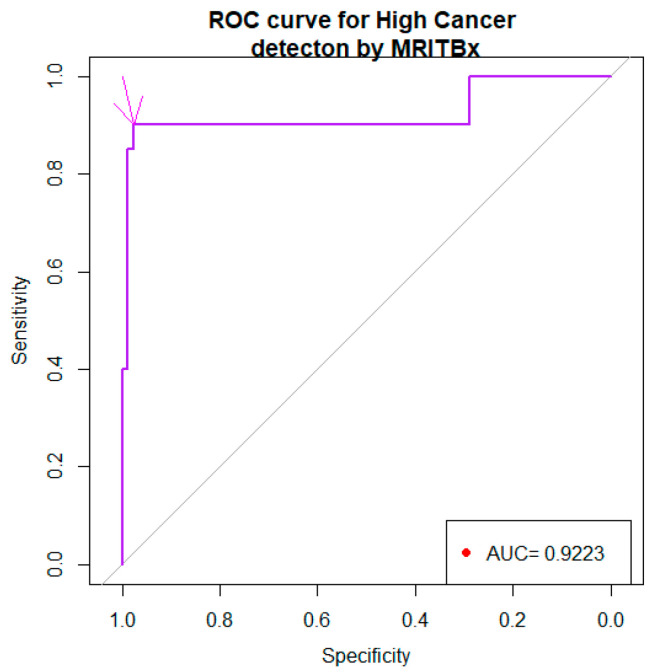
ROC curve of the biomarker (based on MRITBx and other predictors).

**Figure 5 bioengineering-10-01395-f005:**
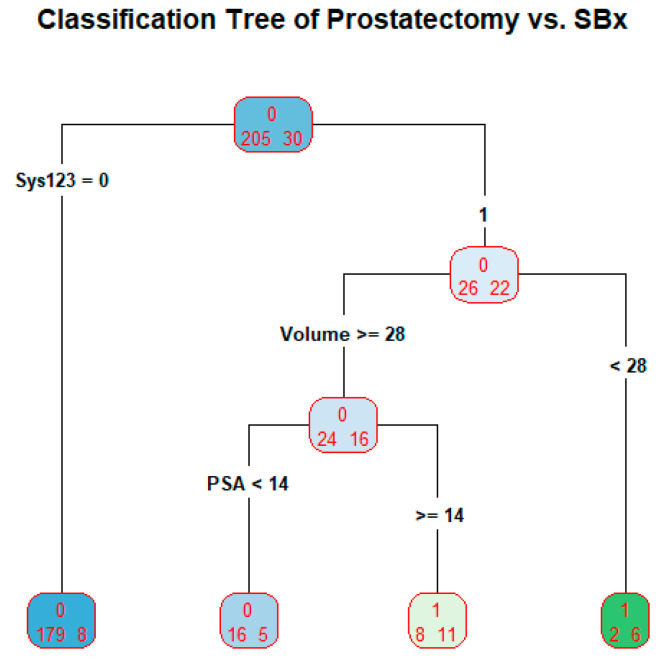
Classification Tree of Prostatectomy vs. SBx.

**Figure 6 bioengineering-10-01395-f006:**
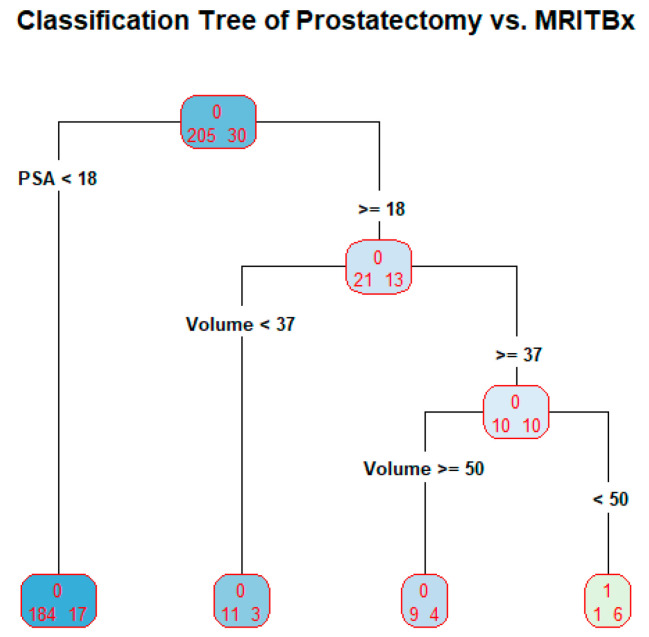
Classification Tree of prostatectomy vs. MRITBx.

**Table 1 bioengineering-10-01395-t001:** Gleason Scores and Grade Groups.

Risk Group	Gleason Grade	Gleason Score
Low/Very Low	Grade 1	Gleason Score ≤ 6
Intermediate (Favorable/Unfavorable)	Grade 2	Gleason Score 7 (3 + 4)
	Grade 3	Gleason Score 7 (4 + 3)
High/Very High	Grade 4	Gleason Score 8
	Grade 5	Gleason Score 9–10

**Table 2 bioengineering-10-01395-t002:** Gleason Grades by Biopsy.

Grades	SBx	Prostatectomy	MRITBx	Prostatectomy
1	41	4	24	3
2	103	133	48	60
3	43	68	11	32
4	20	6	5	3
5	28	24	16	6
Total	235	235	104	104

**Table 3 bioengineering-10-01395-t003:** Cross tabulation of SBx versus prostatectomy.

Prostatectomy (True Condition)	Diagnosed Condition by SBx		Marginal
	Grade ≥ 2	Grade = 1	
Grade ≥ 2	192	38	231
Grade = 1	2	2	4
Marginal	194	41	235

**Table 4 bioengineering-10-01395-t004:** Cross tabulation of MRITBx versus prostatectomy.

Prostatectomy (True Condition)	Diagnosed Condition by MRITBx		Marginal
	Grade ≥ 2	Grade = 1	
Grade ≥ 2	79	22	101
Grade = 1	1	2	3
Marginal	80	24	104

**Table 5 bioengineering-10-01395-t005:** Summary Statistics of the biomarker (based on SBx and other predictors) by the levels of Hcpros.

Levels of Hcpros	Min	I Quartile	Mean	Median	III Quartile	Max
1	−3.413	−0.933	0.0838	−0.693	0.201	1.079
0	−16.776	−3.842	−2.934	−3.205	−2.789	1.756

**Table 6 bioengineering-10-01395-t006:** Summary Statistics of the biomarker (based on MRITBx and other predictors) by the levels of Hcpros.

Levels of Hcpros	Min	I Quartile	Mean	Median	III Quartile	Max
1	−3.811	−1.137	0.499	−0.559	0.3428	1.288
0	−24.531	−19.515	−4.045	−8.649	−3.353	1.463

## Data Availability

Data is available on request.

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
