# Peer review of "Reliability of Systematic and Targeted Biopsies versus Prostatectomy"

_bioengineering, 2023, doi:10.3390/bioengineering10121395_

Round 1

Reviewer 1 Report (Previous Reviewer 1)

Comments and Suggestions for Authors

Can be accepted for publication.

Author Response

Thank you for your comments. We appreciate your input.

Reviewer 2 Report (Previous Reviewer 2)

Comments and Suggestions for Authors

Nothing specific further

Author Response

We have improved the material in the results section as well as in the conclusions section. Please review the new manuscript. We are hoping that revised manuscript meets your approval. Thank you for your input.

This manuscript is a resubmission of an earlier submission. The following is a list of the peer review reports and author responses from that submission.

Round 1

Reviewer 1 Report

Comments and Suggestions for Authors

Dear Author 

This manuscript is about Reliability of Systematic and Targeted Biopsies versus Prostatectomy in prostate cancer patients and well designed .

Can be accepted for publication .

1. What is the main question addressed by the research? Prediction Gleason Grades using classification in three methodology for prostate cancer. 2. Do you consider the topic original or relevant in the field? Does it address a specific gap in the field? Yes    3. What does it add to the subject area compared with other published material? Comparing three methods to choose a appropriate method for prostate cancer diagnosis .   4. What specific improvements should the authors consider regarding the methodology? What further controls should be considered? They can use from combined method for prostate cancer diagnosis (Molecular and Imaging).   5. Are the conclusions consistent with the evidence and arguments presented and do they address the main question posed? Yes   6. Are the references appropriate? Yes   7. Please include any additional comments on the tables and figures.

Well designed and practical manuscript .

Author Response

Dear editors and reviewers,

Thank you so much for your time to review my paper titled “Reliability of Systematic and Targeted Biopsies versus Prostatectomy”. We revised our paper according to all the comments from 3 referees and editors. I used tracking function in word, all revisions to the manuscript was marked using different color, such that any changes can be easily reviewed by you. The following is a cover letter to explain, point by point, the details of the revisions to the manuscript and my responses to the referees’ comments. (My response are in blue color)

  1. Reviewer # 1

Comments and Suggestions for Authors

Dear Author

This manuscript is about Reliability of Systematic and Targeted Biopsies versus Prostatectomy in prostate cancer patients and well designed . Can be accepted for publication.

My response:

Thank you for all the suggestions and comments. I really appreciate your time.

Reviewer 2 Report

Comments and Suggestions for Authors

Overall, the manuscript is not much intriguing. The article needs a thorough improvement so that it can appeal to readers who would be mostly clinicians or biologists. Testing and validation of the predicted model is lacking.

Some minor comments:

Structured abstract doesn’t need headings for each sections.

Table 2, there is a mistake in calculation for SBx. Kindly correct.

Comments on the Quality of English Language

Minor improvement possible.

Author Response

Dear editors and reviewers,

Dear Reviewer,

Thank you so much for your time to review my paper titled “Reliability of Systematic and Targeted Biopsies versus Prostatectomy”. We revised our paper according to all the comments from 3 referees and editors. I used tracking function in word, all revisions to the manuscript were marked using different color, such that any changes can be easily reviewed by you. The following is a cover letter to explain, point by point, the details of the revisions to the manuscript and my responses to the referees’ comments. (My response are in blue color)

  1. Reviewer # 2

Comments and Suggestions for Authors

Overall, the manuscript is not much intriguing. The article needs a thorough improvement so that it can appeal to readers who would be mostly clinicians or biologists. Testing and validation of the predicted model is lacking.

My response:

Thank you for all the suggestions and comments. We have employed the cross - validation method (LOOCV: Leave One Out Cross-Validation) on the logistic regression model contrasting High Cancers vs. Not High Cancers based on Age, Race, Volume, PSA and High Cancers vs. Not High Cancers as per SBx has accuracy 84%. The cross-validation method corroborates the accuracy with its own accuracy calculation of 86%.

We improved our introduction and conclusions to highlight the usefulness of our findings. Please review the new submission.

Some minor comments:

Structured abstract doesn’t need headings for each sections.

My response:

As suggested, I removed the headings in the abstract.

Table 2, there is a mistake in calculation for SBx. Kindly correct.

My response:

I checked our original data; the error is in the count number of Grade 3 under SBx.  We corrected it. The total sample size is 235 now. Thank you so much.

Comments on the Quality of English Language

Minor improvement possible.

My response:

I added substantial material to the manuscript, so that clarity is enhanced.

Reviewer 3 Report

Comments and Suggestions for Authors

1. Highlight literature gaps and provide details of contribution end of the introduction section

2. Add a few more recent literature to support the document strongly 

3.  To enhance the quality of the presentation of work carried out by researchers a workflow diagram can be added in the methodology section along with a discussion.

4. The relations in the conclusion section may be removed. Only the conclusion and significance of the method may be highlighted instead. Finally, it may end with future clinical benefits.

5. Provide all the abbreviations in full forms such as SBx 

Author Response

Dear reviewer,

Thank you so much for your time to review my paper titled “Reliability of Systematic and Targeted Biopsies versus Prostatectomy”. We revised our paper according to all the comments from 3 referees and editors. I used tracking function in word, all revisions to the manuscript was marked using different color, such that any changes can be easily reviewed by you. The following is a cover letter to explain, point by point, the details of the revisions to the manuscript and my responses to the referees’ comments. (My response is in blue color)

Reviewer #3

Comments and Suggestions for Authors

  1. Highlight literature gaps and provide details of contribution end of the introduction section

My response:

We highlighted a gap in the literature and addressed the gap in the introduction section.

  1. Add a few more recent literature to support the document strongly 

My response:

We added 12 recent papers to describe different methods of prostate cancer detection in introduction and conclusion sections. These papers introduce novel detection methods but fail to address how reliable the methods are vis-a-vis prostatectomy. The reason, understandably, could be lack of data on prostatectomy.

  1. To enhance the quality of the presentation of work carried out by researchers a workflow diagram can be added in the methodology section along with a discussion.

My response:

I added the following workflow diagram in methodology section:

  1. Grade 1,2,3,4 or 5, as per SBx in comparison of Grade 1,2,3,4 or 5 as per Prostatectomy (Cohen’s Kappa)
  2. Grade 1,2,3,4 or 5, as per MRITBx in comparison of Grade 1,2,3,4 or 5 as per Prostatectomy (Cohen’s Kappa)
  3. Grade 1 vs. Grade 2,3,4 or 5 as per SBx in comparison of Grade 1 vs. Grade 2,3,4 or 5 as per Prostatectomy (Logistic Regression)
  4. Grade 1 vs. Grade 2,3,4 or 5 as per MRITBx in comparison of Grade 1 vs. Grade 2,3,4,5 as per Prostatectomy (Logistic Regression)
  5. Grade 1or 2 vs. Grade 3,4,5 as per SBx in comparison of Grade 1 or 2 vs. Grade 3,4,5 as per Prostatectomy (Logistic Regression)
  6. Grade 1or 2 vs. Grade 3,4,5 as per MRITBx in comparison of Grade 1 or 2 vs. Grade 3,4,5 as per Prostatectomy (Logistic Regression)
  7. Grade 1or 2 vs. Grade 3,4,5 as per MRITBx in comparison of Grade 1 or 2 vs. Grade 3,4,5 as per Prostatectomy (Classification Tree)

  1. The relations in the conclusion section may be removed. Only the conclusion and significance of the method may be highlighted instead. Finally, it may end with future clinical benefits.

My response:

I added a paragraph on future benefits in the conclusion section at the end.

  1. Provide all the abbreviations in full forms such as SBx 

My response:

A summary of Acronyms is provided at the end of the Introduction. Please review the updated manuscript.

Thank you again for your time

Round 2

Reviewer 2 Report

Comments and Suggestions for Authors

It is still not up to the mark mainly because of the way of writing and lack of systematic validation of the predicted model and should be rejected.

Comments on the Quality of English Language

NIL